# Computational Understanding of Delithiation, Overlithiation, and Transport Properties in Disordered Cubic Rock-Salt Type Li_2_TiS_3_

**DOI:** 10.3390/nano13233013

**Published:** 2023-11-24

**Authors:** Riccardo Rocca, Mauro Francesco Sgroi, Maddalena D’amore, Nello Li Pira, Anna Maria Ferrari

**Affiliations:** 1Department of Chemistry and NIS, University of Turin, 10125 Torino, Italy; maurofrancesco.sgroi@unito.it (M.F.S.); maddalena.damore@unito.it (M.D.); 2Centro Ricerche FIAT S.C.pA, 10043 Orbassano, Italy

**Keywords:** DFT, Li-ion batteries, CRYSTAL, delithiation, diffusion coefficient, Raman

## Abstract

Lithium–titanium–sulfur cathodes have gained attention because of their unique properties and have been studied for their application in lithium-ion batteries. They offer different advantages such as lower cost, higher safety, and higher energy density with respect to commonly adopted transition metal oxides. Moreover, this family of compounds is free from critical raw materials such as cobalt and nickel. For cathode materials, a crucial aspect is evaluating the evolution and behavior of the structure and properties during the cycling process, which means simulating the system under lithium extraction and insertion. Structural optimization, electronic band structures, density of states, and Raman spectra were simulated, looking for fingerprints and peculiar aspects related to the delithiation and overlithiation process. Lithium transport properties were also investigated through the nudged elastic band methodology. This allowed us to evaluate the diffusion coefficient of lithium, which is a crucial parameter for cathode performance evaluation.

## 1. Introduction

The compositions of cathodes for lithium-ion batteries for electric vehicle applications encompass a complex range of materials that have been extensively used in different applications. Lithium cobalt oxide (LCO) is one of the earliest and most widely used cathode materials. It offers high energy density and a good cycle life. Lithium nickel manganese cobalt oxide (NMC) has gained significant popularity due to its balanced performance and can offer a good compromise between energy density, power capability, and cycle life. Different NMC compositions (for example, NMC111, NMC532, NMC622, NMC811) with varying nickel, manganese, and cobalt ratios are available, allowing customization based on specific application requirements. Lithium iron phosphate (LFP) cathodes are known for their excellent safety and long cycle life. LFP has a lower energy density compared to other cathode chemistries but is widely used in applications where safety and longevity are crucial. Lithium nickel cobalt aluminum oxide (NCA) offers high energy density and good power capability. Lithium manganese oxide (LMO) cathodes are often employed in low-cost consumer electronics due to their low cost and good thermal stability. However, they have lower energy density and may exhibit a shorter cycle life than other cathode chemistries. Most of the cathode materials cited above contain cobalt, which is expensive and has ethical and environmental concerns associated with its mining and supply chain [1,2,3,4].

Lithium–titanium–sulfur (LTS) cathodes have gained attention as potential alternatives to traditional cathodes in lithium-ion batteries. This new class of cathodes offers certain advantages and has been studied for its unique properties. First, the abundance and cost: titanium and sulfur are relatively abundant and low-cost compared to cobalt and nickel, which is a key advantage for large-scale battery production. LTS cathodes have shown improved safety characteristics compared to some cobalt-containing cathodes, because sulfur is less prone to thermal runaway reactions, contributing to the improvement of battery safety. LTS cathodes can potentially offer higher energy density compared to traditional NMC cathodes, leading to longer-lasting batteries. While LTS cathodes offer advantages such as cost, safety, and energy density, further research and development are needed to overcome their challenges and optimize their performance for practical application in lithium-ion batteries.

Lithium–titanium–sulfur compounds can be found with different stoichiometry ratios and crystal structures; among them, the Li_2_TiS_3_ cubic rock salt-type has generated considerable attention due to its high capacity and better performance shown by the analogue layered monoclinic structure [5,6]. Its structure and electronic and spectroscopic properties were widely investigated in our previous work [7], where a strong relationship between structure and properties was highlighted. A further step is to evaluate the system evolution during the phenomena occurring during cycling, since the cathode undergoes lithium extraction and insertion. Generally, cathode materials are affected only by delithiation, but for this system, is reported that a certain percentage of extra lithium atoms can be accepted during discharge, reaching an overlithiation state like Li_2+x_TiS_3_ [8,9,10].

In this paper, different states of delithiation and overlithiation of Li_2_TiS_3_ have been simulated with a DFT approach and analyzed in detail by considering the redox processes involved during charge and discharge and the variations in the electron and lattice structures. Raman spectra have been also computed for delithiated and overlithiated Li_2_TiS_3_ in order to identify spectroscopic features correlated with different structures as specific fingerprints. The evaluation of the energetics involved in Li addition and extraction has been also used to estimate the open-circuit voltage (OCV) versus state of charge (SOC) curve, in order to predict and rationalize the cathode performances. The transport properties of lithium were explored, yielding an estimate of the diffusion coefficient.

## 2. Computational Details

Calculations for structure optimization and electronic and spectroscopic properties were performed employing the periodic CRYSTAL17 code [11], based on DFT Hamiltonians and localized Gaussian-type basis functions. The B3LYP [12,13] “hybrid” functional was chosen in combination with split-valence triple-zeta basis sets plus polarization (TZVP). The basis set consisted of Ahlrichs TZVP [14] functions, characterized by S with 73211-6111-1 → [5s,4p,1d], Ti with 842111-631-411 → [6s,3p,3d], and Li with 6211-2 → [4s,1p] [15]. In the CRYSTAL code, the truncation criteria of the Coulomb and exchange infinite lattice series are controlled by five thresholds, which were set to 7 (T1–T4) and 14 (T5). The SCF convergence threshold for the energy was set to 10^−8^ Hartree for the structural optimization, whereas for vibrational frequency calculations, it was set to 10^−10^. The reciprocal space sampling was based on a regular Pack–Monkhorst sub-lattice grid centered at the Γ point, with shrinking factors of 6 and 12 along each of the reciprocal lattice vectors, which corresponds to a number of k-points in the irreducible part of the first Brillouin zone that varies from 6 for more symmetric spatial groups to 12 for systems with low symmetry. Raman spectra, simulated with the harmonic approximation, were computed with well-assessed tools available in the CRYSTAL code [11]. For electron band calculations, the sampling of the first Brillouin zone of the reciprocal space was performed with reference to the Bilbao Crystallographic Server [16,17] and to those reported in the relevant literature [18]. The k-points grid was built as described above. 

Calculations for the evaluation of lithium transport properties were performed using the Quantum Espresso ab initio simulation package [19,20]. The exchange–correlation interaction between electrons was described using the generalized gradient approximation (GGA) with the Perdew–Burke–Ernzerhof exchange–correlation functional optimized for solids (PBEsol) [21]. Frozen-core all-electron calculations were made possible by using the projector-augmented wave (PAW) method [22]. The Brillouin zone was sampled using a 2 × 2 × 1 irreducible Monkhorst–Pack k point grid [23] during structural optimizations. The electron wavefunctions were expanded in a plane wave basis with an energy cutoff of 50 Rydberg, and a Gaussian smearing [24] of 0.01 Ry was applied to improve the convergence of the self-consistent field procedure. The tolerance of total energy convergence was set to 1.0 ×  10^−6^ Ry between successive ionic iterations. To ensure the accuracy of the calculations, the convergence threshold for self-consistency during electronic iterations was considered as 1.0 ×  10^−9^ Ry, with a mixing factor of 0.2. The lattice parameters and atomic coordinates were optimized until their residual force was less than 1.0 ×  10^−5^ Ry Å^−1^.

The nudged elastic band [25] methodology was applied thanks to the tool implemented in the Quantum ESPRESSO code. This method was used to find the minimum energy path (MEP) between the initial and final transition states and was previously used to simulate cathodic materials for Li-ion batteries [26]. The resolution of the initial path is defined by the number of images used to construct the path. The climbing image NEB (CI-NEB) method [27], which forces one of the images to be at the maximum of MEP, was employed. The initial and final points of the paths were optimized, and intermediate and transition-state (TS) structures were produced, starting from the optimized one by manually moving the lithium atom within the structure. During the simulation, the energy of each image was calculated to reconstruct the energy curve along the path and find the energy barrier ($\Delta E_{a}$) in both diffusion directions. 

The diffusion coefficient can be obtained according to the dilute diffusion theory [28] by applying the equation:(1)$$D=a^{2}\vartheta *exp\left[-\frac{\Delta E_{a}}{k_{B}T}\right]$$
where *a* is the hop distance in cm, ϑ* the effective hopping frequency, *k_B_* the Boltzmann constant, and *T* the temperature. Estimation of the effective frequency can be carried out with two simple phonon calculations [29]. It is defined as the ratio of the product of the *N* normal frequencies of the entire system in the initial equilibrium state to the *N* − 1 normal frequencies of the system constrained in the saddle point configuration:(2)$$\vartheta *=\frac{\prod_{j}^{N}\vartheta_{j}}{\prod_{j}^{N-1}\vartheta_{j}^{\prime }}$$

The saddle point phonon calculation is characterized by only one negative frequency, which ensures the correct transition point selection for further calculations. Phonon calculations were performed after a self-consistent calculation with the same computational details as detailed above but with a convergence threshold for self-consistency of 1.0 ×  10^−10^ Ry. The calculations of the spectroscopic properties were carried out with a threshold for self-consistency of 1.0 × 10^−12^ Ry and a mixing factor of 0.3.

Considering the reaction
(3)$${Li}_{x_{1}}TiS_{3}\to {Li}_{x_{2}}TiS_{3}^{(x_{1}-x_{2})-}+\left(x_{1}-x_{2}\right){Li}^{+}$$
the value of the open-circuit voltage (OCV) can be calculated with the following equation reported by Ceder et al. [30]:(4)$$V\left(x_{1},x_{2}\right)\approx \frac{E\left({Li}_{x_{1}}TiS_{3}\right)-E\left({Li}_{x_{2}}TiS_{3}\right)-\left(x_{1}-x_{2}\right)E\left(Li\right)}{\left(x_{1}-x_{2}\right)F}$$
where $E\left({Li}_{x_{1}}TiS_{3}\right)$ and $E\left({Li}_{x_{2}}TiS_{3}\right)$ are the lithiated and delithiated state internal energy, E(Li) is the internal energy of metallic (body-centered cubic) lithium, and *F* is the Faraday constant. For obvious reasons $x_{1}$ must be greater than $x_{2}$.

## 3. Results and Discussion

### 3.1. Pristine System

The structure for Li_2_TiS_3_ was proposed and fully investigated in our previous work [7]. Beginning from the crystallographic experimental data, it was possible to generate a structure capable of reasonably simulating the real system. We considered a primitive cell of 54 atoms: 18 Li, 9 Ti, and 27 S (Li_18_Ti_9_S_27_ with respect to the primitive cell) to reproduce the Li_2_TiS_3_ stoichiometry. Lithium and titanium atoms were arranged in the structure to generate all the possible configurations (4,686,825, which becomes 4023 when considering symmetry degeneracy) and classified by introducing geometrical-order parameters describing the internal organization of Ti atoms in the Li sublattice. We found that all the structures could be classified into two main groups:
Ordered: titanium atoms show a certain degree of nanoclustering, forming clusters, rows, and even planes.Dispersed: characterized by no Ti clustering and containing titanium atoms that are almost uniformly distributed in the Li sublattice.

For a better comprehension of the structures, an example of ordered and dispersed structure is reported in Appendix A. 

The results show that the larger the dispersion of titanium atoms, the more stable the structure, and a strong correlation between structural, electronic, and Raman spectroscopic features was highlighted. The proposed configuration for Li_2_TiS_3_ was indeed a pseudo-cubic structure with Ti dispersed in the lattice with no sign of internal clusterization.

### 3.2. Delithiation

The selected structure for our delithiation study was the most stable one obtained in our previous work [7]. The structure is schematically represented in Figure 1, together with the lithium site label that will be used in the following discussion.

Delithiation occurs according to the following generalized reaction:(5)$${Li}_{2}TiS_{3}\to {Li}_{2-x}TiS_{3}+x{Li}^{+}+xe^{-}$$
with *x* formally ranging from 0 (no delithiation) to 2 (complete delithiation). Taking into account all possible delithiation states, including the extreme condition Li_0_TiS_3_, a total of 262,144 possible configurations can be generated, as reported in Appendix A. However, as reported in the literature [9], Li_2_TiS_3_ can be delithiated down to Li_0.34_TiS_3_, corresponding to a delithiation of 83%, and for this reason, we considered the removal of up to 15 lithium atoms from the Li_18_Ti_9_S_27_ cell.

The energy involved in the process of Li extraction of Equation (5), Δ*E_Extraction_*, was computed according to the following equation:(6)$$\Delta E_{Extraction}=\left(E_{{Li}_{2-x}TiS_{3}}+{xE}_{Li}\right)-E_{{Li}_{2}TiS_{3}}$$
where $E_{{Li}_{2-x}TiS_{3}}$ is the energy of the delithiated structure, *E_Li_* is the lithium energy, *x* is the number of Li atoms removed, and $E_{{Li}_{2}TiS_{3}}$ is the energy of the starting structure without delithiation.

We first considered the case of a single delithiation; in this case, 18 possible delithiated cells can be generated. All possible configurations showed a difference in terms of energy of about 0.2 eV in the worst case, while several structures were almost equivalent, with an energy difference amounting to less than 0.04 eV (see Table 1). 

Considering higher delithiation states, the number of possible configurations increases significantly. For example, at 33% delithiation (six Li atoms removed), the number of configurations rises to 18,564, while at 50%, the maximum of 48,620 configurations is reached. These numbers are not manageable with the DFT method we applied. For this reason, we selected three different possible configurations for each delithiation state. The delithiation sites were selected to have the largest dispersion of vacancies to avoid clusterization of defects. In this way, we believe we have managed to reasonably represent the system during delithiation.

Moreover, we considered a further extraction of a second Li from the most stable mono-delithiated structure (see Table 1). The second lithium atom was removed from three different positions, thus yielding three different configurations, whose energetics are reported in Table 2.

On the basis of Equation (6), the energy required for the second lithium extraction can be retrieved from the following equation (with *x*_1_ > *x*_2_).
(7)$$\Delta E_{Single\;Li}=\left(E_{{Li}_{2-x_{2}}TiS_{3}}+E_{Li}\right)-E_{{Li}_{2-x_{1}}TiS_{3}}$$

As can be seen in Table 3, the required energy for the extraction of the second lithium atom is far less than the first one: the energy required for the mono-delithiation is around 4 eV, whereas the second lithium extraction needs only about 2 eV. 

The differences in energy involved in the mono- and double-delithiation can be rationalized considering the different mechanisms involved in the reaction. In the mono-delithiation case, one electron leaves the structure, causing an electron vacancy that is localized on one or more S^2−^ atoms near the newly formed vacancy. 

In contrast, when two lithium atoms are removed from the structure, two sulfur atoms close to the vacancy are oxidized from S^2−^-S^2−^ to (S-S)^2−^, as reported by Sakuda et al. [10]:(8)$${Li}_{2}TiS_{3}^{2-}\;\to {Li}_{1.77}Ti{\left(S^{2-}\right)}_{2}{\left({S_{2}}^{2-}\right)}_{1}+0.23{Li}^{+}+0.23e^{-}$$

Thus, the removal of a pair of lithium atoms leads to the formation of a S-S bridge, which partially compensates for the cost of lithium extraction. The removal of an even number of lithium atoms causes the formation of S-S bridges, while in odd delithiated structures, an electron hole is generated and localized on sulfur atoms.

Indeed, in Appendix A, the relative stabilities of the three selected configurations for each delithiation state are reported. The configurations with an even number of lithium atoms removed are characterized by energy differences of few hundred meV and are rarely over 1 eV, whereas the odd delithiated configurations show a wider range of energies that can reach up to 3 eV. This proves that the formation of S-S bonds will stabilize the structure and minimize the energy differences between the configurations, while the electron deficiency distributed in the odd delithiated structure yields higher destabilization and increases the energy variability among selected configurations. This phenomenon is even more evident when considering the ΔE_Single Li_ reported in Table 3.

### 3.3. Electronic Properties and Structure

The different mechanisms involving mono- and double-delithiation (and more generally even and odd numbers of Li atoms removed) can be monitored by focusing on the electronic properties of the resulting structures. The projected densities of states (PDOS) are reported in Figure 2, where projections for different sulfur atoms near to and far from the Li vacancies were considered.

In the mono-delithiated structure, the electron hole is localized in the 3p orbitals of a pair of adjacent sulfurs that bear an approximate spin density of 0.5 |e|, and the overall system becomes spin-polarized. The hole formation is monitored using PDOS, with a large upshift of the S 3p states close to the top of the valence bands that move in the gap along the minority spin channel (β) and a significant downshift of 3p states from the conduction bands. The reduced formal charge on these sulfur couples reduces the electrostatic repulsion, leading to a significant contraction of their mutual distance by about 0.3 Å. The stability of the S couple is favored by the presence of titanium (beside lithium) cations in the first shell of neighbors surrounding the vacancy, since titanium atoms, bearing a larger electron charge with respect to lithium atoms, can provide better electrostatic stabilization. When further Li is extracted, regardless of the site of the structure, the second electron hole causes the complete reduction of the S couple to (S-S)^2−^ with the formation of a new S-S covalent bond (the S-S distance changes from an average value of 3.3 Å to 2.3 Å). The S-S bond formation happens with every two Li atoms removed: this means that one bond is formed when two Li are removed, two bonds when four lithiums are extracted and so on, up to seven bonds in the final configuration with a fourteen-lithium removal.

The PDOSs show that the same two sulfur atoms in double-delithiated structures are characterized by different contributions to the density of states with respect to the mono-delithiated case. The impurity states in the middle of the gap are recovered, the spin polarization is lost, and the new feature at 4.75 eV appears that indicates the formation of the reduced S-S couple.

The effects of electron deficiency and the vacancy formation highly influence the band gap. High variability with the lithiation state can be highlighted: in the fully lithiated structure, a band gap of 2.46 eV is reported [7], while proceeding with the delithiation, a decrease is observed down to 2.15 eV when 14 lithium atoms are removed (see Appendix A). In contrast, when an odd number of lithium atoms are removed, the band gap changes from direct to indirect and reduces significantly to 1.7 eV for 15 Li removed, due to the presence of defect states in the gap along the β channel. Examples of band structures can be found in Appendix A.

Changes in the oxidation state and the resulting formation of S-S bonds have a significant effect on internal organization of the Li_2_TiS_3_ structure; indeed, proceeding with lithium extraction, the initial cubic order tends to disappear, and a more amorphous organization can be observed. The amorphization is due to the shift of sulfur atoms from their lattice positions to form the S-S bonds; this causes a deformation of the ideal octahedron structure, as bonds tend to elongate or shorten based on their relative position with respect to the vacancy sites and S-S bonds. The deviation from the cubic arrangement towards a more amorphous structure increases with the amount of delithiation, as expected. In Figure 3, four different structures with a different number of lithium atoms removed are reported: 2, 4, 6, and 8. During the delithiation process an expansion of the cell can be observed. The volume expands from 948 Å^3^ for the fully lithiated structure to 1143 Å^3^ for the fully delithiated structure.

### 3.4. Overlithiated Structures

As mentioned at the beginning, the Li_2_TiS_3_ system undergoes overlithiation, accepting lithium atoms beyond the stoichiometric ratio. The maximum lithium content reported in the literature is Li_2.23_TiS_3_, and considering our cell with 18 lithium atoms, this corresponds to the addition of two extra lithiums. The process follows the reaction
(9)$${Li}_{2}TiS_{3}+x{Li}^{+}\to \;{Li}_{2+x}TiS_{3}^{x+}$$

The first step for the calculation of the energy involved in overlithiation was to find the most stable configuration for the mono-overlithiated structure; the extra lithium atom was inserted in different positions. Considering a cube where the vertices are occupied by Li, Ti, or S atoms, three possible configurations were explored: on the cube face, on the edge, and at the center (as highlighted in Figure 4a). Following the geometry optimization, when a lithium atom is inserted on the cube face or edge, it slowly moves to occupy an interstitial position in the tetrahedron defined by four cations (see Figure 4b).

A key aspect to be evaluated is the relative Ti/Li composition of the tetrahedron around the extra atom inserted. For the mono-overlithiated structure, three possible configurations were considered:Three Li and one Ti (case 1);Two Li and two Ti (cases 2–4);Four Li (case 3).

The other two possible configurations (three Ti and one Li, four Ti) were not possible because of the low number of Ti atoms and the dispersion that characterize the structure. As highlighted in Table 4, the preferred configuration is the one with only Li atoms as first neighbors (case 3), followed by case 1. This suggests that the more Li atoms present in the first sphere of coordination of the interstitial lithium, the more stable the structure. This finding can be easily related to the small steric hindrance and electrostatic repulsion exerted by Li^+^ with respect to Ti^4+^.

The same behavior can be observed when two additional lithium atoms are considered. In this case, four different cases were analyzed:Case 1: one lithium atom added in the center of a tetrahedral site formed by Li atoms, the other one surrounded by three Li atoms and one Ti;Case 2: one lithium atom added in the center of a tetrahedral site formed by Li atoms, the other surrounded by two Ti and two Li;Case 3: one lithium atom in the center of a tetrahedral site formed by three Li and one Ti, the other one surrounded by four Li;Case 4: one lithium atom in the center of a tetrahedral site formed by three Li and one Ti, the other one surrounded by two Ti and two Li;

As expected, in this case, tetrahedrons with the highest percentage of Li atoms are also more suitable for the insertion of the interstitial Li.

From the energetic point of view, the overlithiation can be assumed as a favored process, as evident from ΔE_Extraction_ computed according to Equation (6) and ΔE_Single Li_ according to Equation (7), reported in Table 4. This means that insertion of an extra Li produces an energy gain of 1.60 eV and of 1.43 eV for a second Li insertion. 

The insertion of one or two Li yields reduction of one or two Ti atoms from Ti^4+^ to Ti^3+^, as monitored using PDOS and reported in Figure 5. 

### 3.5. Raman Spectra

We simulated Raman spectra to identify the presence of features that could be used as fingerprints of specific chemical bindings. From the spectra reported in Figure 6, the region between 450 and 500 cm^−1^ can be assigned to the S-S bond symmetric stretching, as reported in the literature [8,32,33]. In more detail, in the case of a double remotion of Li atoms in the former structure, a peak falling at 479 cm^−1^ is clear and, thanks to the analysis of movements along the mode, it can be mostly attributed to the stretching of the disulfur bridge. The peak is absent in the case of single delithiation, as is evident in the bottom panel of Figure 6. Depending on the delithiation site, the location, and the number of S-S bridges in the crystal structure, the peaks assigned to the stretching modes can shift to lower or higher frequencies, as is clear in the spectra reported in the lower panels of Figure 6. Below 300 cm^−1^, we can identify modes assigned to Li-S stretching in the bulk. Above this threshold, the involvement of Ti-S stretching (symmetric or antisymmetric) rises, and in the range between 350 and 450 cm^−1^, the Ti-S bond stretching modes are coupled with Li-S stretching. In the meantime, the coupling between different Ti-S stretching modes becomes stronger at higher frequencies. Thanks to the Raman technique, it is possible to identify the presence or absence of disulfur bridges in delithiated structures. Going beyond the stoichiometric structure to the Li-rich side of LTS composition, also involving the presence of Ti^3+^ species (as testified by the electron structure analysis), we can identify a feature centered at 394 cm^−1^ in the Raman spectrum coming from a Li-Li stretching. Less intense Raman peaks appear at higher wavenumbers as a result of the coupling of Li-Li stretching with that of Li-S.

### 3.6. SOC-OCV

Based on the energy data reported before, it is possible to simulate the state of charge (SOC) versus open-circuit voltage (OCV), which is an important indicator of the overall cell performances. From this curve, it is possible to understand how the OCV, expressed in Volts, changes as a function of the SOC, expressed in percentage, where 0% corresponds to a fully discharged cell and 100% to a fully charged cell. The value of the OCV can be calculated with Equation (4), reported by Ceder et al. [30].

The calculation of the SOC-OCV curve must be performed by considering both delithiated and overlithiated structures. If only delithiated structures are taken into account, misleading results are obtained. In fact, as shown in Figure 7, the voltage profile would be between 4.3 and 3 V, with an inverse trend with respect to real cathode materials (lower voltage at lower state of charge). However, considering the most overlithiated structure as the starting point for the calculation, an SOC-OCV curve between 1.4 and 2.8 V is obtained, in good agreement with the literature data that report a working voltage between 1.4 and 3 V [8,9]. During charge, the first lithium atoms to leave the structure are the ones related to the overlithiation, since characterized by the lowest extraction energy (see Table 2 and Table 4). This phenomenon is accompanied by the oxidation of titanium atoms from 3+ to 4+. Once the stoichiometric conditions are reached, after around 20% SOC, further lithium removal determines the oxidation of sulfur from S^2−^-S^2−^ to (S-S)^2−^, as highlighted in the delithiation section. Fluctuations in the curve derive from the fact that odd and even delithiation involving different mechanisms lead to different extraction energies (see Table 4) and different V (x_1_, x_2_) values.

### 3.7. Lithium Diffusion

We now move to the exploitation of the lithium transport properties in order to evaluate the diffusion coefficient. For pseudo-cubic disordered systems, two possible hopping mechanisms, both involving a Li vacancy, are reported in the literature [34,35,36] and are sketched in Figure 8: “direct mechanism”: the lithium atom moves along a straight line from the initial position to the final position (the vacancy site), highlighted in red in Figure 8;“indirect mechanism”: the lithium atom first moves to an interstitial position in the center of the cube formed by Li and Ti atoms, then migrates to the final Li vacancy position, highlighted in blue in Figure 8.

In Table 5, the energy barriers, the effective frequencies, and the relative diffusion coefficient computed at 300 K for systems containing only one vacancy are reported. For each mechanism, three different paths were considered (labelled from 1 to 3 in Table 5) by changing the initial Li sites and the final vacancy sites. Clearly, each scenario is characterized by two activation energies (forward and backward processes) and, consequently, by two diffusion coefficients since the two directions are not equivalent. Hop distances *a* are reported in Table 5, and, since the structure is pseudo-cubic and slightly distorted, the distances travelled by the lithium atoms are different from each other even considering the same diffusion mechanism. The diffusion coefficient was computed according to Equation (1) and is reported in Table 5.

The fluctuation of the diffusion coefficient values can be associated with the variability of neighbors around the initial and final sites since, as highlighted before, the system stability is highly influenced by the local atomic arrangement. From the analysis of the diffusion paths it is possible to assess that the more lithium atoms are present in the environment around the initial or final configurations, the more stable are the involved structures and, consequently, the migration is preferred toward those configurations. The diffusion coefficient obtained by these types of calculations ranges from 6.38 × 10^−10^ to 1.24 × 10^−6^, which is in line with the data reported in the literature: 4.5 × 10^−8^–5.3 × 10^−7^ cm^2^/s [37].

## 4. Conclusions

In this work, disordered cubic Li_2_TiS_3_ was investigated, focusing on aspects related to lithium extraction, insertion, and transport. The system was investigated by means of periodic DFT calculations for the simulation of structural, electronic, and spectroscopic properties. The initial system was already studied in our previous work [7], and the most stable configuration was selected as the starting point for this study.

Initial analysis of the possible delithiation state and the consequent number of possible configurations was carried out. Due to the large number of possible configurations, a selection of representative structures was performed (three different delithiation configurations) to reduce the calculation effort to an affordable amount. From the results obtained, the delithiation level strongly influences the structure evolution: each time a pair of lithium atoms are removed, one S-S bond is formed, formally corresponding to the oxidation of a S^2−^-S^2−^ couple into a (S-S)^2−^ one; in delithiation involving an odd number of lithium atoms, an electron vacancy is formed and localized on a pair of adjacent sulfur atoms.

During the delithiation process, the structure undergoes gradual amorphization and the cell volume expands. This system is also characterized by the possibility of accommodating a certain percentage of extra lithium atoms in the crystalline structure, which means that more energy can be stored in and gathered from the battery. The Li insertion is energetically favored, and extra Li is interstitially hosted in tetrahedron sites preferentially surrounded by other Li cations. 

From the data derived from the delithiation and overlithiation processes, it was possible to evaluate the SOC/OCV curve, which represents the evolution of voltage with the state of charge of the cell. The results obtained are in line with the experimental ones reported in the literature, confirming the accuracy of the adopted method. Moreover, overlithiated structures were found to be crucial for the correct simulation of SOC/OCV curves. Raman spectroscopy allows us to discriminate between structures with a different lithium content. Specific peaks can be assigned to S-S bonds, helping us to recognize delithiated structures.

To investigate the transport properties of lithium, diffusion coefficients were evaluated. Cubic structures are characterized by two diffusion paths, with different environments near the involved lithium atom that highly influence the motion of the lithium. However, the values are in line with those reported in the literature.

## Figures and Tables

**Figure 1 nanomaterials-13-03013-f001:**
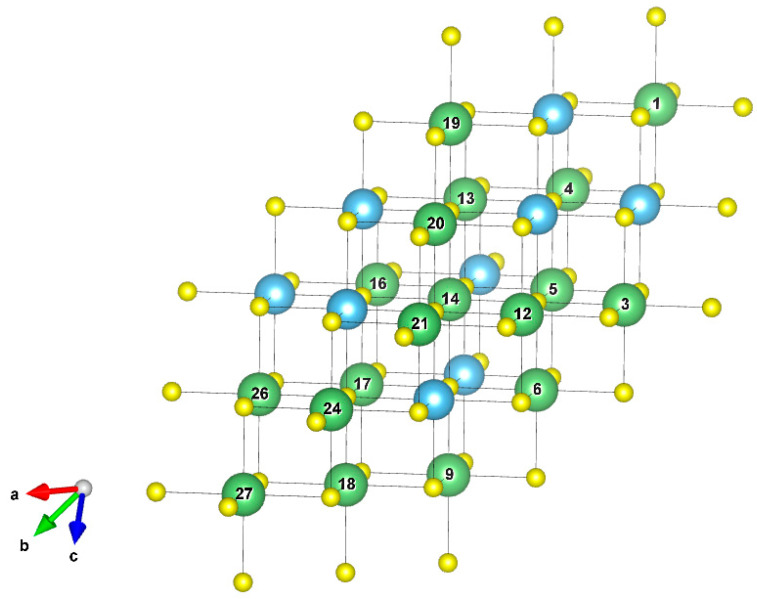
Lithium site numbering. Sulfur is shown in yellow, titanium in light blue, and lithium in green. Structures visualized and figures produced with the help of VESTA software (ver. 3.5.8) [31].

**Figure 2 nanomaterials-13-03013-f002:**
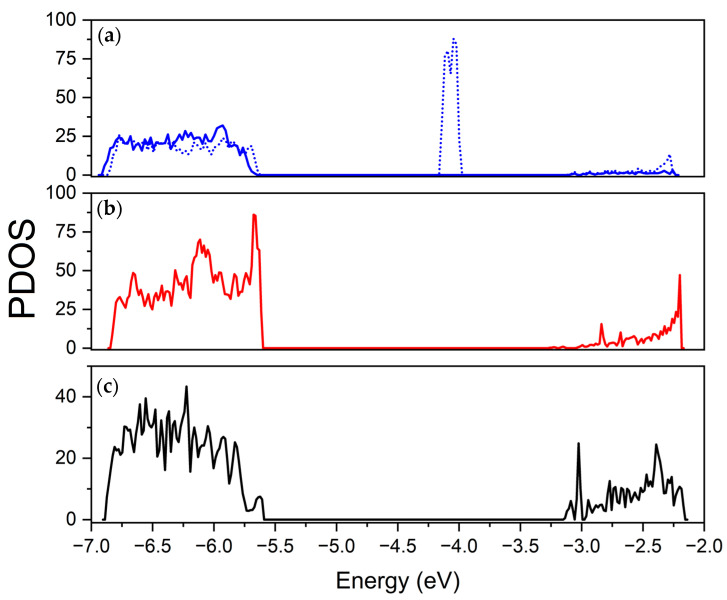
Projected densities of states for different S atoms: (**a**) S atom near the vacancy in a mono-delithiated structure; (**b**) S atom in a double-delithiated structure in the same position as the mono-delithiated one; (**c**) S atom in non-defective structure. In the mono-delithiated case, the α and β contributions are visualized with the solid and dotted line, respectively.

**Figure 3 nanomaterials-13-03013-f003:**
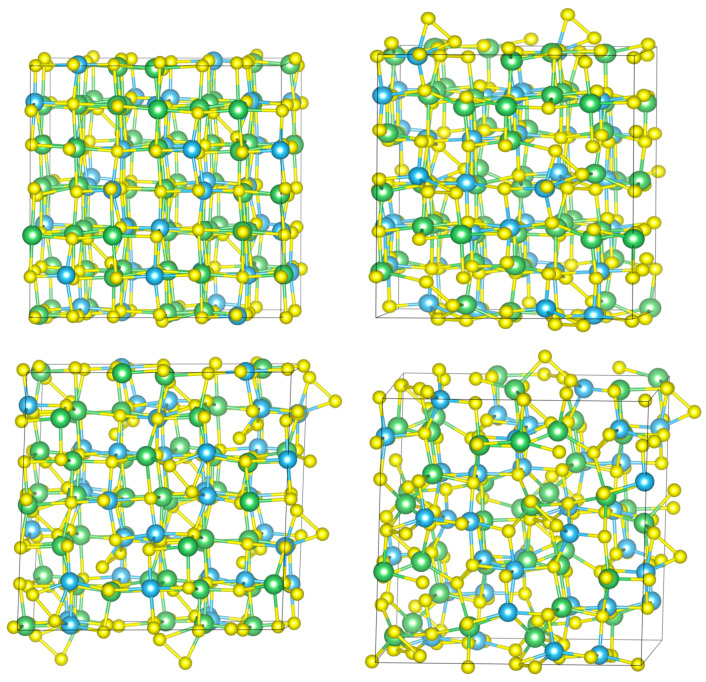
Structures with a different number of lithium atoms removed: top left, 2; top right, 4; bottom left, 6; and bottom right, 8. The structures reported are converted into conventional cells, resulting in a visualization of four times the unit cell. Sulfur is shown in yellow, titanium in light blue, and lithium in green. Structures visualized and figures produced with the help of VESTA software [31].

**Figure 4 nanomaterials-13-03013-f004:**
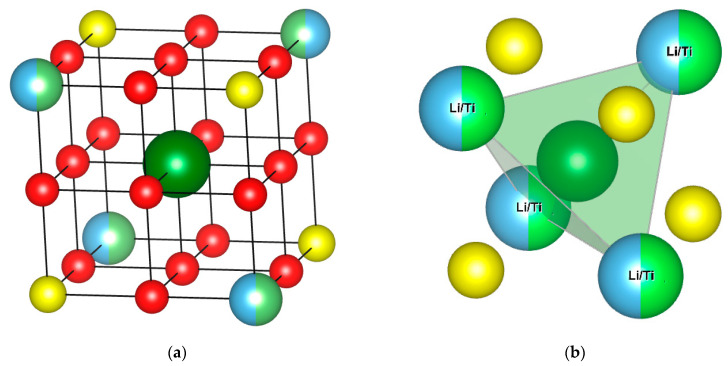
(**a**) The tested sites are in red and dark green is the most stable configuration. (**b**) Schematics of the tetrahedron formed by Li and Ti atoms (the most stable configuration is in dark green). Sulfur is shown in yellow, titanium in light blue and lithium in green. Structures visualized and figures produced with the help of VESTA software [31].

**Figure 5 nanomaterials-13-03013-f005:**
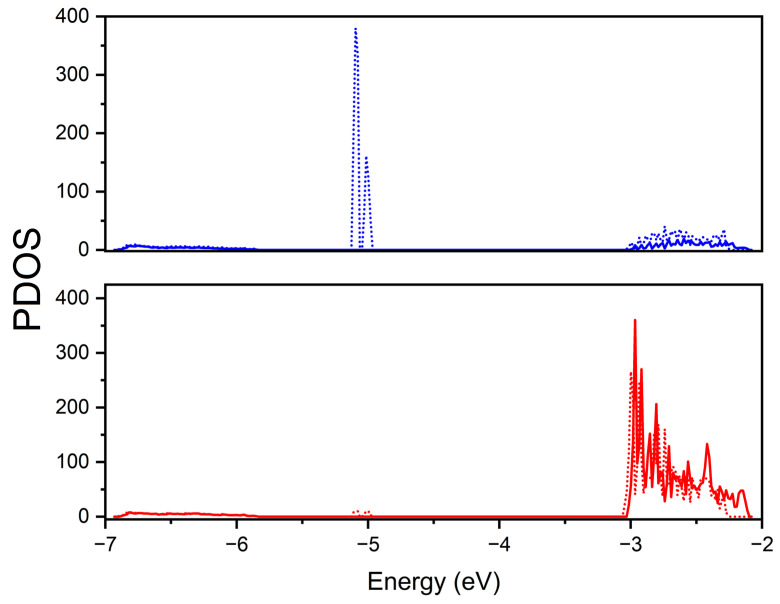
Projected density of state of titanium atoms for double-overlithiated structure. Ti^3+^ on top and Ti^4+^ on bottom. The α and β contributions are visualized with the solid and dotted line, respectively.

**Figure 6 nanomaterials-13-03013-f006:**
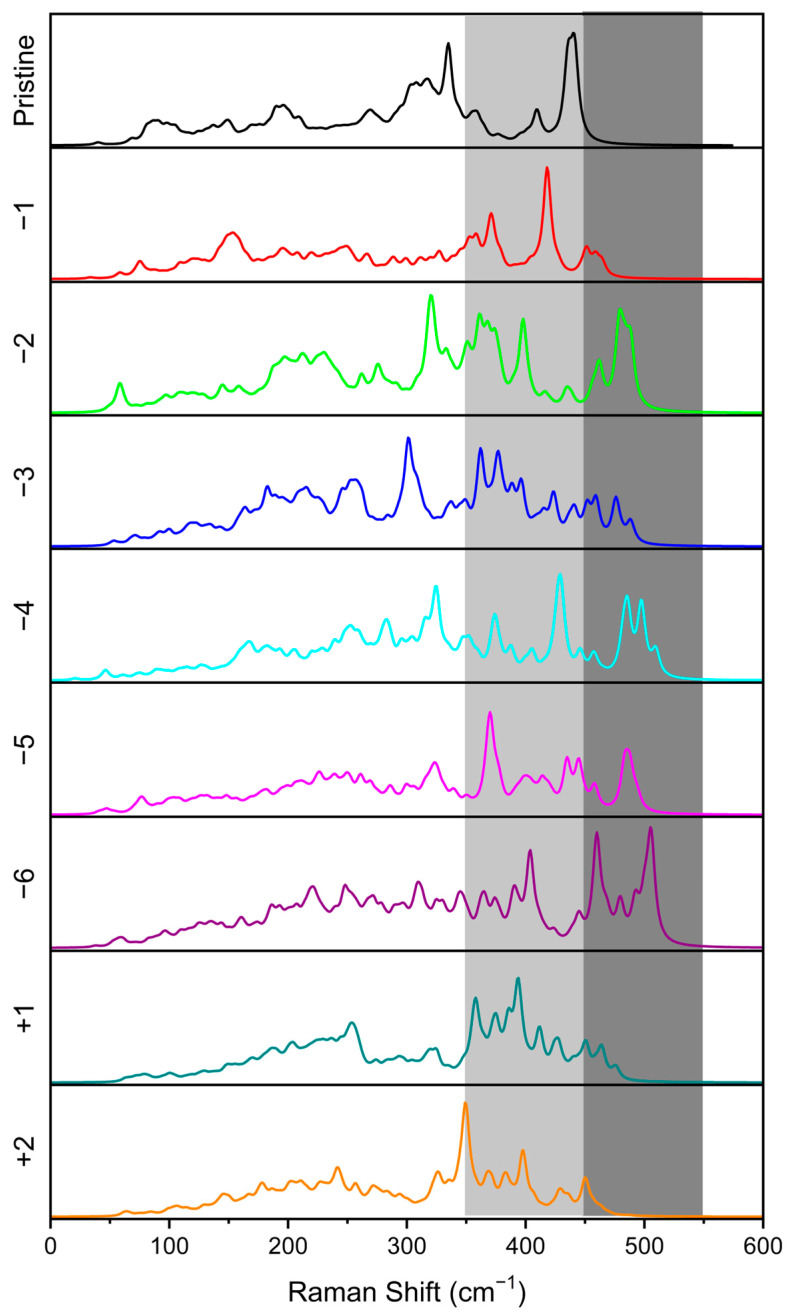
Simulated Raman spectra for delithiated and overlithiated structures. At the top, the Raman spectrum of pristine LTS is reported. From top to bottom, the delithiation rises, from 1 to 6 lithium atoms removed. The two bottom spectra are related to the two overlithiated structures. The intensities are normalized to 1 for a better comparison. The region where Ti-S bond stretching occurs is highlighted in light gray, while dark gray is the region characterized by S-S symmetric stretching.

**Figure 7 nanomaterials-13-03013-f007:**
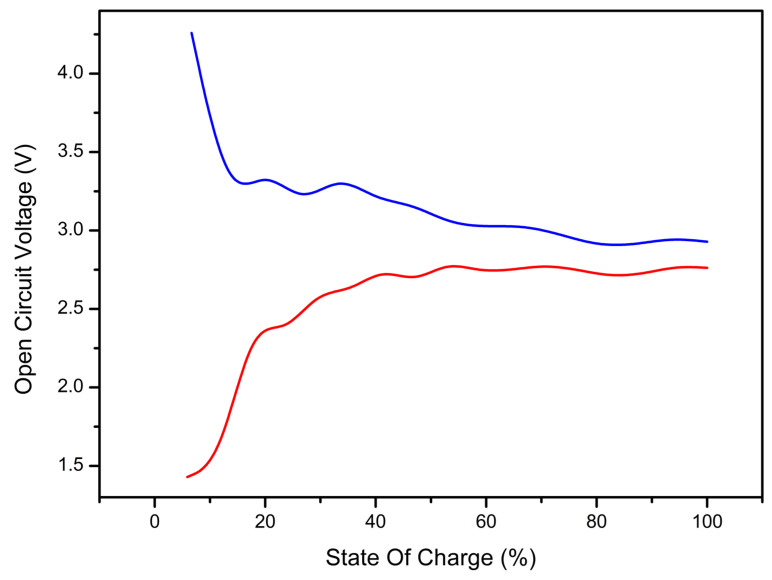
State of charge vs. open-circuit voltage curve for the LTS system. Blue is the calculated curve considering the fully lithiated structure as the starting point. Red is the curve obtained by considering the overlithiated structure as the starting point.

**Figure 8 nanomaterials-13-03013-f008:**
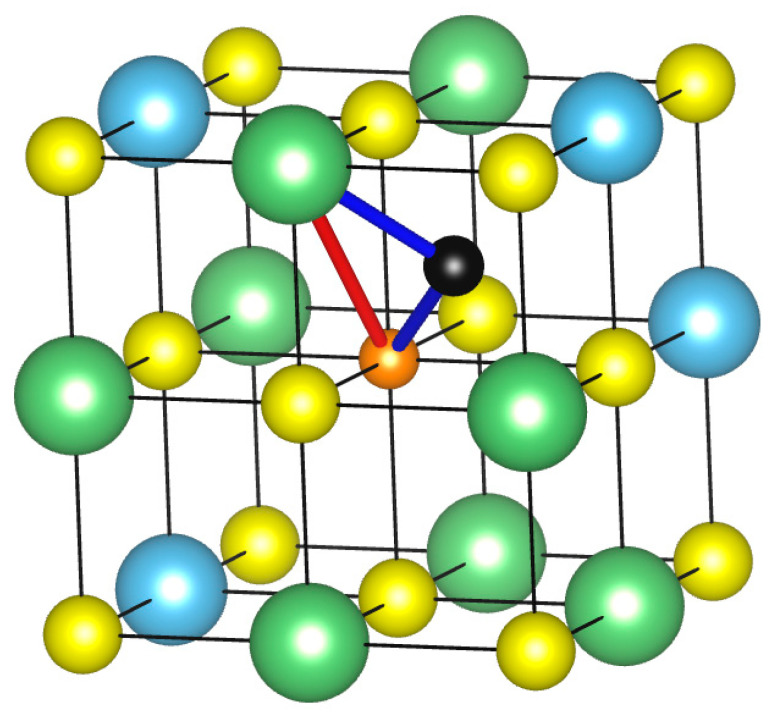
Graphical visualization of two possible migration paths in cubic structures: direct and indirect (highlighted in red and blue, respectively). The interstitial position is shown in black and the vacancy in orange. Sulfur is shown in yellow, titanium in light blue, and lithium in green. Structures visualized and figures produced with the help of VESTA software [22].

**Table 1 nanomaterials-13-03013-t001:** Energies of the 18 possible configurations for the monovacancy system. The stability is relative to the most stable one. ΔE_Extraction_ was computed according to Equation (6).

Vacancy Position	ΔE_Extraction_ (eV)	Relative Stability (eV)
**20**	4.18	0.000
**13**	4.19	0.011
**21**	4.21	0.035
**16**	4.22	0.037
**5**	4.25	0.067
**9**	4.26	0.082
**27**	4.28	0.106
**12**	4.30	0.120
**14**	4.31	0.129
**1**	4.32	0.143
**26**	4.32	0.146
**19**	4.33	0.148
**3**	4.33	0.150
**4**	4.33	0.151
**18**	4.33	0.154
**17**	4.34	0.157
**24**	4.35	0.175
**6**	4.40	0.223

**Table 2 nanomaterials-13-03013-t002:** Reaction energy for delithiation.

	ΔE_Extraction_ (eV)	ΔE_Single Li_ (eV)
**Mono-delithiated**	4.18	
**Double-delithiated 1**	6.172	1.991
**Double-delithiated 2**	6.251	2.070
**Double-delithiated 3**	6.511	2.331

**Table 3 nanomaterials-13-03013-t003:** Reaction energy for the most stable structure for each delithiation state. x is the number of Li atoms removed. Also reported is the required energy for a single lithium extraction calculated considering the most stable structure for each delithiation state.

x	ΔE_Extraction_ (eV)	ΔE_Single Li_ (eV)
1	4.18	4.18
2	6.31	2.14
3	10.22	3.90
4	12.63	2.41
5	16.29	3.66
6	18.30	2.01
7	22.18	3.00
8	24.35	2.17
9	27.21	2.86
10	30.33	3.12
11	32.75	2.42
12	34.87	2.12
13	37.80	2.93
14	41.34	3.54
15	43.93	2.60

**Table 4 nanomaterials-13-03013-t004:** Energy differences for overlithiated structures for the two cases of one and two extra lithium added. The reported energy is relative to the most stable one.

One Li Added			Two Li Added			
	Delta (eV)	ΔE_Extraction_ (eV)		Delta (eV)	ΔE_Extraction_ (eV)	ΔE_Single Li_ (eV)
**Case 3**	0.000	1.605	**Case 1**	0.000	3.034	1.429
**Case 1**	0.056	1.549	**Case 2**	0.280	2.754	1.150
**Case 2**	0.739	0.865	**Case 3**	0.208	2.825	1.221
**Case 4**	0.740	0.865	**Case 4**	1.111	1.922	0.318

**Table 5 nanomaterials-13-03013-t005:** Diffusion coefficient in Li_2_TiS_3_ calculated according to Equation (1). The effective frequency ϑ* in s^−1^ and the activation energy in eV for the two possible diffusion mechanisms (Equation (2)). For each case, three different paths were simulated, and for each one, forward and backward processes are reported. The value of a (hop distance) is reported following the NEB calculation with Quantum ESPRESSO.

	Effective Frequency (s^−1^)	Activation Energy (eV)	Diffusion Coefficient (cm^2^/s)	Hop Distance *a* (Å)
**Direct 1**	3.13 × 10^13^	0.333	1.21 × 10^−8^	1.25
		0.307	3.39 × 10^−8^	
**Direct 2**	3.32 × 10^14^	0.350	6.93 × 10^−8^	1.25
		0.275	1.24 × 10^−6^	
**Direct 3**	1.22 × 10^14^	0.451	6.38 × 10^−10^	1.40
		0.411	2.95 × 10^−9^	
**Indirect 1**	3.97 × 10^13^	0.356	7.09 × 10^−9^	1.31
		0.393	1.70 × 10^−9^	
**Indirect 2**	2.75 × 10^13^	0.291	5.02 × 10^−8^	1.18
		0.259	1.71 × 10^−7^	
**Indirect 3**	6.51 × 10^13^	0.397	3.35 × 10^−9^	1.56
		0.359	1.44 × 10^−8^	

## Data Availability

The data presented in this study are available in the manuscript and the Appendix A.

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
