# Peer review of "Computational Understanding of Delithiation, Overlithiation, and Transport Properties in Disordered Cubic Rock-Salt Type Li2TiS3"

_nanomaterials, 2023, doi:10.3390/nano13233013_

Round 1
Reviewer 1 Report
Comments and Suggestions for Authors
This paper presents a systematic simulation study on the disordered cubic rock-salt type 3 Li2TiS3. I recommend the paper to be accepted after the following issues are well addressed.
1. How to define ‘disordered’? As the authors mentioned in Page 4 Line 155, there is some correlation between the degree of disorder and the simulation result. How did the authors cope with this issue exactly?
2. Did the authors find any valence change of the Ti atoms in the simulation? Further, the statement in Page 15, Line 389-391 should be elaborated in more details.
3. How would the structure of Li2TiS3 change if it is delithiated/lithiated for cycles? Would the structure maintain?
4. English need to be further improved. For instance, Page 9, Line 276 ‘bonds tends to’; Page 13, Line 344, ‘Raman spectra to possible identify the’. Etc.
Comments on the Quality of English LanguageMinor editing of English language required
Author Response
- How to define ‘disordered’? As the authors mentioned in Page 4 Line 155, there is some correlation between the degree of disorder and the simulation result. How did the authors cope with this issue exactly?
We agree with the referee that the term ‘disordered’ can be misleading, the structures studied in our previous work (https://doi.org/10.3390/nano12111832) are all characterized by a certain level of disorder. We used improperly the word ‘disordered’ to describe the structures characterized by titanium nanoclustering, and we think that the more appropriated word is ‘dispersed’, that indicates more precisely the situation of Titanium atoms uniformly distributed in the crystal lattice. As presented in our previous work, we were able to discriminate between ordered and dispersed structure by introducing two geometrical descriptors based on the distances and dispersion of Ti atoms. We modified the sentence in Page 4 Lines 151-152 accordingly:
“Dispersed: characterized by no Ti clustering and containing titanium atoms that are almost uniformly distributed into the Li sublattice.”
- Did the authors find any valence change of the Ti atoms in the simulation? Further, the statement in Page 15, Line 389-391 should be elaborated in more details.
We modified the text according to the referee suggestions, the valence change of Ti atoms is highlighted in the overlithiation section Page 11 Lines 329-330. We further elaborated the sentence highlighted in Page 14, Lines 379-384:
“During charge, the first lithium atoms to leave the structure are the one related to the overlithiation, since characterized by the lowest extraction energy (see Table 2 and 4). This phenomenon is accompained by the oxidation of titanium atoms from 3+ to 4+. Once the stoichiometric conditions are reached, after around 20% SOC, further lithium removal determines the oxidation of sulfur from S2--S2- to (S-S)2-, as highlighted in the delithiation section.”
- How would the structure of Li2TiS3 change if it is delithiated/lithiated for cycles? Would the structure maintain?
In the literature is reported that the phenomenon is reversible (DOI:10.1038/s41598-018-33518-4, https://doi.org/10.3390/ma15093037). The scope of this paper is to deeply evaluate the mechanisms that characterize the delithiation and overlithiation processes. The structure evolution under cycling will be investigated in future works.
- English need to be further improved. For instance, Page 9, Line 276 ‘bonds tends to’; Page 13, Line 344, ‘Raman spectra to possible identify the’. Etc
We carefully checked the manuscript for the English improvement, changes have been highlighted in the revised document. We are grateful to the referee suggestions.
Reviewer 2 Report
Comments and Suggestions for Authors
The submitted article provides detail about the simulated behavior of a Li cathode material for battery applications, composed of earth abundant Li, Ti, an S elements. The authors have conducted a DFT simulation of the lithiation/delithiation process in this material, with an explanation of the atomic and structural changes in the material. The article is significant and worthwhile, but needs editing for clarity and syntax.
Comments on the Quality of English LanguageThe authors would benefit from editing by a native speaker of English. There are many minor grammatical and syntax based changes that could be made to improve the article. The number of instances are too high to cite completely, but several examples are provided throughout the article for changes. The entire article is advised to be examined for these improvements.
Line 33-35. This sentence could be broken into two sentences at the point: “… Lithium Iron Phosphate (LFP) cathodes are known for their excellent safety and long cycle life. LFP has a lower energy density compared to other cathode chemistries but is widely used in applications where safety and longevity are crucial.”
Line 49. “improved” rather than improve.
Line 66. “detail”, not “details”
Line 246-248. Sentence is confusing as written. Please consider an edit for clarity
Line 263. The phrase “highly also” appears awkward.
Line 289. Undergo should be “undergoes”
Line 303-305. The sentence is unclear. The reference to figure 4b does not appear to easily relate to figure 4a. Is 4b a subset of the face or edge of 4a? It seems that there should be a way to better present the investigated interstitial sites.
Lines 417-420. Syntax is awkward, and the sentences seem to be running onto each other.
Line 437. “spectroscopic” rather than “spectroscopical” The latter is not a valid word.
Line 455. Correct the initial sentence. From the data for delithiation and overlithiation”, it” was possible to evaluate the SOC/OCV curve, …
Author Response
Line 33-35. This sentence could be broken into two sentences at the point: “… Lithium Iron Phosphate (LFP) cathodes are known for their excellent safety and long cycle life. LFP has a lower energy density compared to other cathode chemistries but is widely used in applications where safety and longevity are crucial.”
Line 49. “improved” rather than improve.
Line 66. “detail”, not “details”
Line 246-248. Sentence is confusing as written. Please consider an edit for clarity
Line 263. The phrase “highly also” appears awkward.
Line 289. Undergo should be “undergoes”
Line 303-305. The sentence is unclear. The reference to figure 4b does not appear to easily relate to figure 4a. Is 4b a subset of the face or edge of 4a? It seems that there should be a way to better present the investigated interstitial sites.
Lines 417-420. Syntax is awkward, and the sentences seem to be running onto each other.
Line 437. “spectroscopic” rather than “spectroscopical” The latter is not a valid word.
Line 455. Correct the initial sentence. From the data for delithiation and overlithiation”, it” was possible to evaluate the SOC/OCV curve, …
We are grateful for the referee suggestions that helped us to improve the English grammar and syntax, we modified the manuscript accordingly. The changes are highlighted in the revised document. Figure 4 was modified to present in the best way the investigated interstitial sites.
Reviewer 3 Report
Comments and Suggestions for Authors
The study successfully employed simulations to investigate structural optimization, electronic band structures, density of states, and Raman spectra in the context of Li2TiS3. This comprehensive analysis aimed at identifying distinctive features and characteristics associated with the delithiation and overlithiation processes. Furthermore, the examination of lithium transport properties using the Nudged Elastic Band methodology adds a valuable dimension to our understanding of the material. Given the depth and relevance of the findings, I believe the manuscript is well-prepared for publication, meeting the standards for contribution to the field.
Comments on the Quality of English LanguageMinor editing of English language required
Author Response
We are grateful for the referee suggestion to carrefully revise the manuscript, this helped us to improve the English grammar and syntax, we modified the manuscript accordingly.
Reviewer 4 Report
Comments and Suggestions for Authors
Rocca et al., studied, the delithiation, overlithiation, and transport properties in disordered cubic rock-salt type Li2TiS3 using computation approach. The manuscript can be of interest to readers of Nanomaterials and can be considered for publication in Nanomaterials following a minor revision.
1. The authors should provide a color detail about atoms in Figure 1, Figure 3, and Figure 8.
2. In Figures 2 and 5, the Y-axis label is missing.
3. Please prove the fermi energy difference for Figures 2 and 5.
Author Response
- The authors should provide a color detail about atoms in Figure 1, Figure 3, and Figure 8.
The Figures’ captions were modified according to the referee suggestions.
- In Figures 2 and 5, the Y-axis label is missing.
The Figures were modified according to the referee suggestion.
- Please prove the fermi energy difference for Figures 2 and 5.
It is well known that the fermi level depends on the electron contents and the number of atoms in the system, and as expected in our structures we observed a slightly downshift of the fermi level reducing the lithium content and a slightly upshift increasing the number of atoms.